# Unravelling the Role of Cancer Cell-Derived Extracellular Vesicles in Muscle Atrophy, Lipolysis, and Cancer-Associated Cachexia

**DOI:** 10.3390/cells12222598

**Published:** 2023-11-09

**Authors:** Akbar L. Marzan, Sai V. Chitti

**Affiliations:** Department of Biochemistry, La Trobe Institute for Molecular Science, La Trobe University, Melbourne, VIC 3086, Australia; a.marzan@latrobe.edu.au

**Keywords:** cancer-associated cachexia, extracellular vesicles, muscle atrophy, lipolysis, browning

## Abstract

Cancer-associated cachexia is a metabolic syndrome that causes significant reduction in whole-body weight due to excessive loss of muscle mass accompanied by loss of fat mass. Reduced food intake and several metabolic abnormalities, such as increased energy expenditure, excessive catabolism, and inflammation, are known to drive cachexia. It is well documented that cancer cells secrete EVs in abundance which can be easily taken up by the recipient cell. The cargo biomolecules carried by the EVs have the potential to alter the signalling pathways and function of the recipient cells. EV cargo includes proteins, nucleic acids, lipids, and metabolites. Tumour-secreted EVs have been found to alter the metabolic and biological functions of adipose and muscle tissue, which aids in the development of the cachexia phenotype. To date, no medical intervention or FDA-approved drug exists that can completely reverse cachexia. Therefore, understanding how cancer-derived EVs contribute to the onset and progression of cancer-associated cachexia may help with the identification of new biomarkers as well as provide access to novel treatment alternatives. The goal of this review article is to discuss the most recent research on cancer-derived EVs and their function in cellular crosstalk that promotes catabolism in muscle and adipose tissue during cancer-induced cachexia.

## 1. Cancer-Associated Cachexia

Cancer-associated cachexia is a complex multifactorial wasting syndrome that affects up to 80% of patients with advanced cancer [1,2]. Around 9 million patients around the world suffer from this phenomenon every year [3]. Depletion of skeletal muscle and adipose tissue and ongoing weight loss are some of the major hallmarks of cancer-associated cachexia [4]. The underlying mechanisms of the loss of host muscle and fat mass induced by the tumour are often multifactorial and complex [5]. Several seminal studies suggest that various pro-cachectic factors, such as proteolysis-inducing factor (PIF), myostatin, and activin A; inflammatory cytokines, such as interleukin 1, 6, and 10; and interferon gamma secreted by the primary tumour drive pro-atrophic signals and promote substantial metabolic changes. This primarily leads to loss in skeletal muscle mass and fat mass, irreversible by conventional nutritional support [6]. Noteworthily, it is strongly associated with poor patient prognosis, decreased functional activity, and reduced quality of life [7,8]. It also weakens patients to an extent that neither chemotherapy nor surgery can be tolerated [5,9]. Overall, cancer-associated cachexia is known to account for approximately one third of all cancer-related deaths worldwide [9]. 

Based on the degree of recduction in body mass index, energy stores, and the extent of ongoing weight loss, cancer-associated cachexia is classified into three stages: pre-cachexia, cachexia, and refractory cachexia. Patients with less than 5% weight loss, along with anorexia and metabolic changes, are considered to be in the pre-cachexia stage. This stage is often followed by cachexia, where patients undergo weight loss of more than 5% within a span of six months and body mass index (BMI) is also known to reduce beyond 20 kg/m^2^, along with reduced food intake and increased systemic inflammation. Finally, patients reach the refractory stage of cachexia where the chance of survival becomes as little as three months, owing to higher levels of active catabolism. Weight loss management at this stage is no longer possible and the patients stop responding to any treatment [1]. 

## 2. Skeletal Muscle Wasting during Cancer-Associated Cachexia

Skeletal muscle wasting and atrophy is a prominent feature of cancer-associated cachexia [10,11]. In a healthy individual, skeletal muscle comprises approximately 30–40% of the total body mass and contributes to maintaining metabolic homeostasis of the body [12,13]. These muscles not only serve as a reservoir of the body’s amino acids needed for protein synthesis and energy production, but also act as a primary site for glucose uptake and storage [14]. Therefore, maintenance of muscle protein homeostasis is crucial for healthy muscle function and the regulation of energy–protein balance throughout the body [15]. Abnormalities in protein synthesis, protein degradation, amino acid metabolism, apoptosis, and an impaired capacity for regeneration together are known to contribute to muscle wasting [16]. Muscle wasting is also characterized by a substantial loss of myofibrillar proteins, decrease in muscle fibre cross-sectional area, and myonuclear number [17]. Cancer cells secrete various pro-inflammatory and pro-cachectic factors which tend to wreck the balance between the anabolism and catabolism of proteins, thus resulting in negative energy–protein balance and muscle loss [18,19]. Along with reduced protein synthesis, cancer-induced muscle wasting is mediated by the activation of the ubiquitin–proteasome system (UPS), lysosome-dependent degradation, apoptosis, and autophagy, leading to increased protein degradation [20,21,22] (Figure 1A).

Circulating tumour-derived factors activate various intracellular signalling pathways that mediate muscle protein degradation and reduce protein synthesis in the process [23,24]. Inflammatory cytokines, PIF, activin A, and myostatin induce ubiquitin–proteasome degradation and autophagy pathways in muscles via transcription factors such as the forkhead box (FOXO) family [2,21,25]. This leads to the induction of E3 ubiquitin ligases (muscle atrophy F-box protein 1 (MAFbx)/atrogin-1 and the muscle RING finger-containing protein 1 (MuRF1) that promotes muscle protein degradation and wasting [26]). Tumour necrosis factor-alpha (TNF-α) is one of the most prominent pro-inflammatory cytokines that is involved in muscle wasting. It functions by activating NF-κB and ubiquitin E3 ligase pathways to promote muscle protein breakdown [27]. NF-κB has also been found to be activated in cachectic muscles, which dysregulates the expression of paired box7 (PAX7), impairing muscle cell differentiation and repair during cancer-induced muscle atrophy [28]. Interleukin-6 (IL-6) is another well-studied cytokine known to play an important role in cancer cachexia progression. IL-6 has been found to mediate muscle atrophy though the Janus kinase/signal transducers and activators of transcription (JAK/STAT3) signalling pathway [29]. Transforming growth factor-β (TGF-β) and members of its superfamily myostatin and activin have been reported to promote muscle catabolism by stimulating FOXO, which is followed by the upregulation of the UPS pathway in muscle [20]. Furthermore, TNF-like weak inducer of apoptosis (TWEAK) and interferon-γ (INF-γ) were found to activate the p38 mitogen-activated protein kinases (p38-JAK-MAPK) signalling pathway which in turn activates caspases to induce programmed cell death, apoptosis of muscle cells, and promote proteolysis of myofibrillar proteins [21]. In addition to the activation of catabolic pathways, the inhibition of anabolic pathways is reported during cancer-associated muscle wasting. Insulin-like growth factor 1(IGF-1), for example, is known to activate phosphoinositide 3 kinase/Akt/mammalian or mechanistic target of rapamycin mTOR (PI3K-AKT and mTOR) signalling in muscle and promote protein synthesis. However, during cancer-associated cachexia, the level of IGF-1 has been found to decrease, which subsequently reduces protein turnover in the muscle and perturbs muscle mass [23]. Additionally, cytokines released from cancer cells can also phosphorylate and activate peroxisome proliferator-activated receptor-γ co-activator 1α (PGC-1) via p38 MAPK. Cytokine-induced activation of PGC-1 in muscle cells results in increased respiration and expression of genes involved in mitochondrial uncoupling, and energy expenditure [30]. 

Of note, along with muscle catabolism, tumour-secreted factors also impair muscle stem cell function, resulting in a reduction in the regenerative ability of skeletal muscle [31,32]. Hogan and colleagues reported that tumour-derived chemokine ligand 1 (CXCL1) suppresses myogenesis and alters the skeletal muscle immune microenvironment, and these two phenomena together contribute to cancer-associated muscle wasting [33]. CXCL1 has particularly been found to inhibit muscle cell differentiation by promoting satellite cell proliferation and antagonizing cell cycle exit. In addition, CXCL1 stimulates the expansion of neutrophil–macrophage in skeletal muscle, which also leads to the impairment of the regenerative ability of muscle cells [33]. The cytokine-induced degeneration of muscle cells is further known to partake in overall muscle wasting during cancer-associated cachexia [34,35]. The activation of NF-κB by the cytokine TNF-α has also been shown to participate in inhibiting skeletal muscle differentiation by suppressing MyoD mRNA at the post-transcriptional level. Suppression of MyoD expression in muscle inhibits the formation of new myofibers and causes the degeneration of newly formed myotubes [34]. TNF-α and TGF-β cytokines have also been reported to upregulate metal-ion transporter ZRT- and IRT-like protein 14 (ZIP14) in the cachectic muscle, which obstructs muscle-cell differentiation. ZIP14-mediated zinc accumulation in progenitor and differentiated muscle cells results in repression of the expression of MyoD and Mef2, and the loss of myosin heavy chain [35].

## 3. Adipose Tissue Lipolysis and Browning during Cancer-Associated Cachexia

Loss of whole-body fat is another key feature of cancer-associated cachexia. Fat constitutes almost 90% of the total fuel reserve of the body [36]. The role of adipose tissue is to regulate body temperature through thermogenesis and maintain the energy status of the body by storing lipids and supplying free fatty acids to tissues in need of calories [7,37]. Cancer patients undergoing cachexia have been reported to have both increased plasma fatty acids and glycerol concentrations, suggesting increased lipolysis compared to weight-stable cancer patients [38,39]. Loss of adipose tissue often precedes muscle loss and is now therefore considered to be an independent predictor for the survival of cachectic patients [40]. The mechanisms driving the loss of adipose tissue in cancer patients are thought to be multifactorial [41,42]. Activation of lipolysis in white adipose tissue (WAT) is accompanied by a reduced adipogenesis rate and browning of adipose tissue. These, in turn, lead to loss of fat mass during cancer-associated cachexia [43] (Figure 1B). 

Loss of WAT during cancer-associated cachexia is influenced by various mediators, cytokines, and hormones [42,44]. IL-6 secreted from cancer cells has been shown to induce lipolysis. Binding of IL-6 to its receptor IL-6R/gp130 activates the protein kinase A (PKA), G (PKG), and JAK/STAT pathways, which results in the phosphorylation of hormone-sensitive lipase (HSL). HSL then mediates triglyceride hydrolysis and releases free fatty acid and glycerol into the circulation [45,46,47]. LIF, a member of the IL-6 cytokine family, has also been found to induce lipolysis by binding to the LIFR receptor and activating the JAK/STAT pathway [48]. Furthermore, the decreased rate of lipogenesis triggered by peroxisome proliferator-activated receptor-α (PPARα) has also been reported during cancer-associated cachexia by inducing insulin resistance and decreasing or inhibiting glucose transporter 4 (GLUT4), which in turn inhibits glucose transport and lipogenesis [49]. Additionally, TNF-α and IFN-γ also contribute to cancer cachexia-associated adipocyte wasting through reduction in glucose uptake and promoting insulin resistance [50,51]. 

Brown adipose tissue (BAT) is known to play an important role in thermogenesis and energy balance [52,53]. Several studies have implicated the activation of (BAT) in the pathogenesis of cancer-associated cachexia owing to the increased activity of uncoupling protein 1 (UCP-1) [54,55]. Tumour-derived IL-6 and parathyroid hormone-related protein (PTHrP) activate thermogenic gene expression in adipose tissue and increases UCP-1 expression in WAT [43,56]. UCP-1 decouples mitochondrial respiration from ATP synthesis and directs it towards thermogenesis, increasing lipid mobilization and energy expenditure [43,57]. Tumour-derived lipolytic factor zinc-α2-glycoprotein (ZAG) has been found to act as a lipid mobilizing factor and stimulate adipose tissue wasting [58]. Moreover, it was also found to induce adipose tissue browning by promoting the recruitment of peroxisome proliferator-activated receptor γ (PPARγ) to UCP-1, subsequently resulting in an increased expression of UCP-1 [59].

Overall, cancer cachexia research throughout the years has improved our knowledge of the mediators and processes that regulate this metabolic syndrome. However, targeting these various pro-cachectic and pro-inflammatory mediators with anti-cachectic and anti-inflammatory drugs have shown little to no clinical benefit in combatting cancer-associated cachexia [60,61,62]. Most of the drugs that provided protection in pre-clinical models either exhibited relative inconsistency in clinical trials or revealed significant adverse effects [63]. The complex pathophysiology of cancer cachexia syndrome is the fundamental reason for the disparity in pre-clinical-to-clinical transition [64]. Therefore, battling cachexia with a single therapy may not be the most effective form of treatment. From this perspective, therapies involving different combinations might prove to be more effective [64]. Additionally, expanding our knowledge on cancer-derived extracellular vesicles (EVs) and its content might open novel avenues for developing targeted therapies for cancer-associated cachexia, as EVs have recently been found to participate in both muscle and adipose tissue wasting [65,66].

## 4. Overview of Extracellular Vesicles

Extracellular vesicles (EVs) are nano-sized membrane enclosed vesicles produced by all cell types, including cancer cells. EVs are released into the circulation and they transport functional information to distant sites [67]. Extracellular vesicles have been classified into different categories based on their origin, size, and mode of biogenesis [68,69]. EVs can be broadly classified into small and large EVs [70]. Small EVs include exosomes (30–150 nm) and ectosomes or shedding microvesicles (100–1000 nm), and large EVs include apoptotic bodies (1000–5000 nm), migrasomes (500–3000 nm), exophers (3500–4000 nm), and large oncosomes (1000–10,000 nm) [71] (Figure 2).

Exosomes originate from intraluminal vesicles (ILVs) that are generated by inward budding of endosomal membranes as part of multivesicular bodies (MVBs). MVBs either fuse with lysosomes for degradation or with the plasma membrane for them to be secreted as exosomes [72,73]. However, little is known about the mechanism that determines the fate and cargo sorting of MVBs [74]. The main components needed for exosome biogenesis are found in the ESCRT machinery, which consists of endosomal sorting complexes (ESCRT)-0, I, II, and III. They function sequentially to assemble, group, and arrange ubiquitinated proteins into late endosomes [75]. An ESCRT-independent pathway can also produce MVBs, which is known as the ceramide-dependent pathway. During this process, sphingomyelin builds up in lipid microdomains (lipid rafts), where sphingomyelinases transform it into ceramide. The structural imbalance between ceramide-rich domains and the lipid mono-leaflets prevents the membrane from reverse budding [76]. Tetraspanins such as CD81, CD9, CD37, and CD63 and Ras-associated binding (Rab) GTPases such as Rab27 are also known to be involved in the biogenesis and release of exosomes [77,78]. 

Microvesicles or ectosomes are generated by outward budding of the plasma membrane at the specific site followed by fission and the subsequent release of the vesicles into the extracellular space [71,79]. Originally thought to be large EVs (200–1000 nm), ectosomes can also be produced by outward budding and they can fall under the small EV range (<150 nm) [80,81].

Apoptotic bodies are large EVs formed by outward blebbing of the plasma membrane during apoptosis of dying cells [69]. Migrasomes are large vesicles released by migrating cells. Migrasomes are composed of numerous small vesicles found at the tip of retraction fibre which are long tubular strands that migrating cells leave on their way as they move [82,83]. Exophers are recently identified vesicles that remove cellular waste, damaged organelles such as mitochondria, lysosomes, and protein aggregates to maintain homoeostasis of the cells. They are large membrane-bound vesicles distinct from exosomes or apoptotic bodies since they are produced independently of the ESCRT machinery and do not have phosphatidylserine on their surface. The production of exophers increases in proteostasis-impairing conditions and with the suppression of autophagy or protein turnover [84,85,86,87]. Large oncosomes are micrometre-sized cancer cell-derived large EVs that originate from the plasma membrane of amoeboid cancer cells [88]. The definition of these subtypes of Evs are often overlapping and conflicting across various studies, and hence the International Society for Extracellular Vesicles (ISEV) recommends using the term “extracellular vesicle” as a general nomenclature for vesicles released by the cells [70]. 

EV cargo is highly heterogenous and includes proteins, lipids, messenger RNAs (mRNAs), micro RNAs (miRNAs), DNA, and metabolites [89]. These cargo biomolecules are either expressed on the surface or packaged inside the EVs which presumably reflects the dynamic status of the secreting cells [90,91]. The content of the EVs is thought to modulate the signalling of the recipient cells by either transferring their content into the recipient cells directly [92], or by interacting with the extracellular matrix [93]. Interestingly, cancer cells tend to secrete more EVs than non-cancerous cells [94], which are readily available in the circulation and can be easily taken up by the recipient cells. This, in turn, alters the biological function of the recipient cells [95]. Studies have already shown that EVs play a significant role in cancer progression [96], premetastatic niche formation, metastasis [97,98], chemoresistance [99,100], and immune evasion [101]. Moreover, it is now being considered that EVs could also possibly serve as an important mediator in the development of cancer-associated cachexia. This notion is supported by a wide variety of studies showing that cancer cell-derived condition media containing EVs can induce muscle atrophy and lipolysis in vitro [102,103]. In addition, several other seminal studies have demonstrated the role of EVs in promoting muscle and adipose tissue wasting [66,104,105,106], making EVs an ideal candidate to be investigated for their function in the pathogenesis of cancer-associated cachexia.

## 5. Cancer Cell-Derived EV Protein and Muscle Wasting

Research is yet to uncover the mechanism through which tumour or cancer cells communicate with distal muscle and adipose tissue during cancer-associated cachexia. In recent years, EVs have attracted a lot of attention for their intrinsic ability to participate in intracellular communication and involvement in various physiological and pathological conditions [107,108,109]. Tumour-derived EVs have emerged as a novel route through which cancer cells can communicate with remote organs and tissues such as skeletal muscle and adipose tissue and thereby can influence their function (Figure 3 and Table 1) [105,110].

Mu et al. reported that exosomes secreted by osteosarcoma cells contain notch-activating factors such as miRNA199b-5p, miRNA-21, and notch ligands (DLLs, Jagged). These factors activate the notch-dependent signalling in muscle progenitors, which subsequently results in excessive proinflammatory signalling, impaired myogenesis, and mediated muscle atrophy [104]. Furthermore, along with the Notch gene, the expression of proinflammatory factors such as TNF-α was also found to be upregulated. TNF-α can activate Notch by inducing Jagged1 expression, which in turn hampers the function of muscle stem cells and muscle regeneration [104]. Hu and colleagues demonstrated that Lewis lung carcinoma (LLC)-derived EVs mediate dose-dependent muscle catabolism and wasting in cultured myotubes. Mechanistically, LLC-derived EVs contain IL-6 that fuses with C2C12 by interacting with IL-6R on myotubes. The EV-delivered IL-6 then triggers myotube atrophy by activating the STAT3 pathway [105]. STAT3 activation leads to the activation of forkhead box O3 (FOXO3) and atrogin-1, which is a key ubiquitin ligase that drives the process of muscle wasting [105]. Interestingly, IL-6-mediated activation and the phosphorylation of STAT3 enhances glycolytic effect in C26 colon cancer, which in turn amplifies EV secretion that can further contribute to the induction of skeletal muscle and adipose tissue atrophy, leading to the development of cancer cachexia [111].

Cachectic cancer cells such as C26 murine colon cancer cells and LLC cells were found to release heat shock protein (Hsp) 70 and 90, expressing EVs that promote muscle catabolism via activating the TLR4-p38β-MAPK-C/EBPβ catabolic signalling pathway [65]. Activation of p38-mitogen-activated protein kinase (p38-MAPK) by Toll-like receptor 4 (TLR4) results in the activation of the ubiquitin–proteasome pathway (UPP) and autophagy–lysosome pathway (ALP), both of which result in the loss of myofibrillar proteins, muscle strength, and ultimately muscle mass [65]. p38β-MAPK also activates the transcription factor C/EBPβ in muscles, causing upregulation of muscle atrophy marker atrogin-1 and UBR2, which leads to myofiber atrophy and inhibition of muscle regeneration [112]. This was further supported by a study showing that zinc transporter ZIP4 stimulates RAB27B-mediated release of Hsp70- and Hsp90-positive EVs, which in turn activates p38 MAPK-mediated muscle catabolism and promotes pancreatic cancer-associated cachexia [113]. The presence of Hsp70/90 Evs were also found to cause an increase in the expression of autophagy marker LC3 in both in vitro and in vivo models. Additionally, the activation of TLR4 by Hsp70/90 Evs has also been found to be responsible for the systemic rise in the levels of inflammatory cytokines, including TNF-α and IL-6, during cancer cachexia [65].

A study conducted by Zhang et al. further strengthened the role of tumour-secreted EVs in proteolysis [114]. Growth differentiation factor 15 (GDF-15) packed within EVs has been demonstrated to induce myotube atrophy via BCL-3/caspase-3-mediated apoptotic pathways. GDF-15 has been found to be enriched in exosomes derived from cachexia-inducing C26 cells compared to exosomes derived from non-cachectic MC38 cells. Knockdown of GDF-15 in C26 cells significantly reduced the ability of C26-derived exosomes to cause muscle atrophy. In contrast, overexpressing GDF-15 in MC38 cells significantly enhanced muscle atrophy-inducing ability of MC38-derived exosomes [114]. In addition, EVs produced by oesophageal squamous cell carcinoma (ESCC) were reported to induce apoptosis in muscle cells through releasing prolyl 4-hydroxylase subunit beta (P4HB), which activates the ubiquitin-dependent proteolytic pathway. Activation of ubiquitin-dependent proteolytic pathway results in increasing the ubiquitination level of phosphoglycerate dehydrogenase (PHGDH) which downregulates its stability and subsequently downregulates the level of anti-apoptotic protein Bcl-2 [115]. Taken together, these findings highlight that EVs from tumours contain various signalling molecules and proteolysis-inducing factors that lead to muscle mass wasting during cancer-associated cachexia.

**Table 1 cells-12-02598-t001:** Cancer-derived EV contents and their role in cancer-associated cachexia.

EV Content	EV Source	Cancer Type	Function	Reference
**Proteins**				
ZAG	Patient serum MAC16		Adipose tissue browning, lipolysis	[58,59]
HSP70/90	C26 LLC	Colon cancer Lung cancer	Muscle catabolism, systemic inflammation	[65,113]
Adrenomedullin	Patient plasma and Panc-1	Pancreatic cancer	Lipolysis	[66]
Notch-activating factor	K7M2	Osteosarcoma cells	Impaired myogenesis	[104]
IL-6	LLC	Lung cancer	Muscle catabolism, lipolysis	[105]
GDF15	C26	Colon cancer	Muscle atrophy	[114]
P4HB	ESCC	Oesophageal cancer	Muscle wasting	[115]
PTHrP	LLC	Lung cancer	Adipose tissue browning	[116]
ITGB1	Panc-1 Miapaca-2 Capan-2 Patient serum	Pancreatic cancer	Lipolysis	[117]
ITGA6	Panc-1 Miapaca-2 Capan-2 Patient serum	Pancreatic cancer	Lipolysis	[117]
**miRNAs**				
miR-425-3p	NSCLC	Lung cancer	Adipose tissue browning, inhibits proliferation and differentiation of preadipocytes	[118]
miR-146b-5p	HEK293T HCT-116 Patient tissue	Colorectal cancer	Adipose tissue browning	[119]
miR-410-3p	Patient plasma	Gastric cancer	Inhibits adipogenesis	[120]
miR-204-5p	MDMB231 SK-BR-3 4T1 E0771	Breast cancer	Lipolysis, WAT browning	[121]
miR-155.	MCF-7	Breast cancer	Lipolysis, WAT browning	[122]
miR-21	PC1 Panc-2 Miapaca-2 LLC A549	Pancreatic cancer Lung cancer	Muscle catabolism	[123]
miR-125b-1-3p	C26	Colon cancer	Muscle wasting	[124]
miR-195a-5p	C26	Colon cancer	Muscle wasting	[124]

## 6. Cancer-Derived EV Proteins and Lipolysis

In addition to proteolysis and muscle wasting, Evs could also be involved in the crosstalk between tumour cells and adipocytes that causes lipolysis, browning of adipocytes, energy expenditure, and ultimately fat loss during cancer-associated cachexia (Figure 2 and Table 1). A recent study by Sagar et al. revealed that pancreatic cancer (PC)-derived exosomes can promote lipolysis in both human and murine adipose tissue [66]. The levels of the lipolysis-inducing factor adrenomedullin (AM) was found to be higher in PC-derived EVs isolated from patients’ plasma compared to EVs isolated from non-PC control subjects. AM was found to interact with adrenomedullin receptors (ADMRs) on the surface of adipocyte and stimulate the ERK1, ERK2, and p38-MAPK pathways and promote lipolysis. When PC-derived exosomes were internalised through caveolin- or micropinocytosis-mediated endocytosis, exosomal AM modulates its impact by interacting with ADMRs and increasing the expression of lipolysis marker phosphorylated hormone-sensitive lipase (p-HSL) and phosphorylated perilipin1. This lipolytic effect of PC-derived exosomes was abrogated by ADMR blockers or inhibitors of p38 and MAPK [66]. A similar mechanism to muscle wasting is also triggered in adipose tissue by LLC-derived exosomal IL-6, where it fuses with IL-6R on adipocytes and induce lipolysis via activation of STAT3 [105]. Furthermore, EVs released by LLC and C26 tumour cells have also been demonstrated to contain IL-8, which can induce lipolysis via the extracellular IL-8-mediated NF-κB signalling pathway [125].

Hu et al. have recently reported that LLC-derived EVs can induce adipocyte browning and lipolysis in both in vivo and in vitro [116]. LLC-derived EVs transfer PTHrP and activate the protein kinase A (PKA) signalling pathway in 3T3-L1 adipocytes, which in turn results in the phosphorylation of HSL and causes lipolysis. It is also responsible for the browning of adipose tissue by activating UCP-1. Blocking of PTHrP activity in LLC-derived EVs using a neutralizing antibody and by knocking down parathyroid hormone receptor (PTHR) expression abrogated the lipolytic effect of LLC-derived EVs on adipocytes [116]. Another interesting study conducted by Shibata and colleagues found that pancreatic cancer-derived EVs carry adipocyte-targeting integrins, integrin subunit beta 1 (ITGB1) and integrin subunit alpha 6 (ITGA6), that aid in inducing lipolysis during pancreatic cancer-associated cachexia. ITGB1 and ITGA6 present on pancreatic cancer-derived EVs were found to control the EV tropism toward adipocytes where these EVs stimulate lipolysis via the cyclic adenosine monophosphate (cAMP)-PKA)-HSL pathway [117]. Collectively, these studies demonstrate the clear role of tumour-derived EVs in several aspects of adipocyte functions during cancer-associated cachexia.

## 7. EV-Associated MicroRNAs and Cancer-Associated Cachexia

It is of no surprise that different types of tumours or cancer cells produce numerous tumour-derived EVs with a variety of distinct cargoes that are implicated in controlling various biological processes, disease progression, and the metabolic state of the body [126]. There is a growing body of evidence to support the idea that tumour-derived EVs can initiate adipocyte browning, lipolysis, and muscle atrophy during cancer-associated cachexia via noncoding RNA cargo contained within EVs (Figure 3 and Table 1). 

Exosomes secreted by non-small-cell lung tumours (NSCLC) contain miR-425-3p, which has been demonstrated to initiate white adipose tissue browning. Additionally, it has also been shown to inhibit both the proliferation and differentiation of preadipocytes [118]. Adipose tissue browning has also been demonstrated to be induced by exosomal miR-146b-5p that is produced from colorectal cancer cells. Exosomal miR-146b-5p directly binds and represses the downstream gene homeodomain-containing gene C10 (HOXC10), which then stimulates WAT browning and hypermetabolism of lipids, thereby controlling lipolysis [119]. miRNA analysis of subcutaneous adipose tissue and serum exosomes from gastric cancer-associated cachexia patients unfold the upregulation of miR-410-3p expression. miR-410-3p inhibits adipogenesis and lipid accumulation through inhibition or downregulation of insulin receptor substrate 1 (IRS-1) and adipose differentiation factors such as C/EBP-a and PPAR-γ [120]. Exosomal miR-204-5p secreted by breast cancer cells have been reported to promote lipolysis and WAT browning by inducing the hypoxia-mediated leptin signalling pathway [121]. Moreover, breast cancer exosomes have recently been demonstrated to shuttle miR-155 to adipocytes where it can induce WAT browning and lipolysis by suppressing ubiquitin 1 (UBQLN1) [122].

In addition to adipose tissue wasting, miRNAs have also been shown to regulate muscle wasting [123,127]. For instance, miR-21-containing EVs secreted by lung cancer and pancreatic cancer cells can stimulate TLR7 or TLR8 on mouse and human myoblasts, which accelerates muscle catabolism. Activated TLR7/8 causes muscle apoptosis via c-Jun N-terminal kinase, resulting in muscle mass loss or atrophy [123]. Interestingly, C26-secreted exosomes were revealed to be enriched with miR-195a-5p and miR-125b-1-3p by miRNA, which have been found to induce muscle wasting via Bcl-2-mediated apoptosis during colon cancer-associated cachexia [124]. Altogether, these studies provide strong evidence that cancer-derived EVs are enriched with miRNAs that can induce atrophy in both muscle and fatty tissues. EV miRNA research is now considered to be a promising field for advancing our understanding and knowledge of cancer-associated cachexia.

## 8. Future Perspective

It has been well established that cancer cells produce more EVs than healthy cells and the cargo of EVs secreted by the cancerous cells differs from that of non-malignant cells [128,129,130]. Researchers aim to take advantage of this fact to identify circulating EV cargoes that can serve as biomarkers for the early diagnosis of various diseases. The benefits of EVs are their abundance, unique DNA/RNA/protein profiles, and most effective transfer of information in their target cells. The role of EVs in cancer-associated cachexia holds great interest since these nano-sized vesicles have the ability to facilitate communication across various tissues and organs by transporting proteins and miRNAs. Given that cancer-associated cachexia is a systemic metabolic syndrome, EVs play a crucial role in aiding tissue crosstalk. Therefore, improved knowledge about the function of EVs and their contents can be utilized for identifying potential biomarkers involved in cancer-associated cachexia. Additionally, such information can also be employed for therapeutic benefits. This knowledge can also help future researchers in the development of targeted anti-cancer drugs to decrease the release of inflammatory cytokines and atrophy-inducing factors that promotes proteolysis and lipolysis during cancer-induced cachexia. 

## Figures and Tables

**Figure 1 cells-12-02598-f001:**
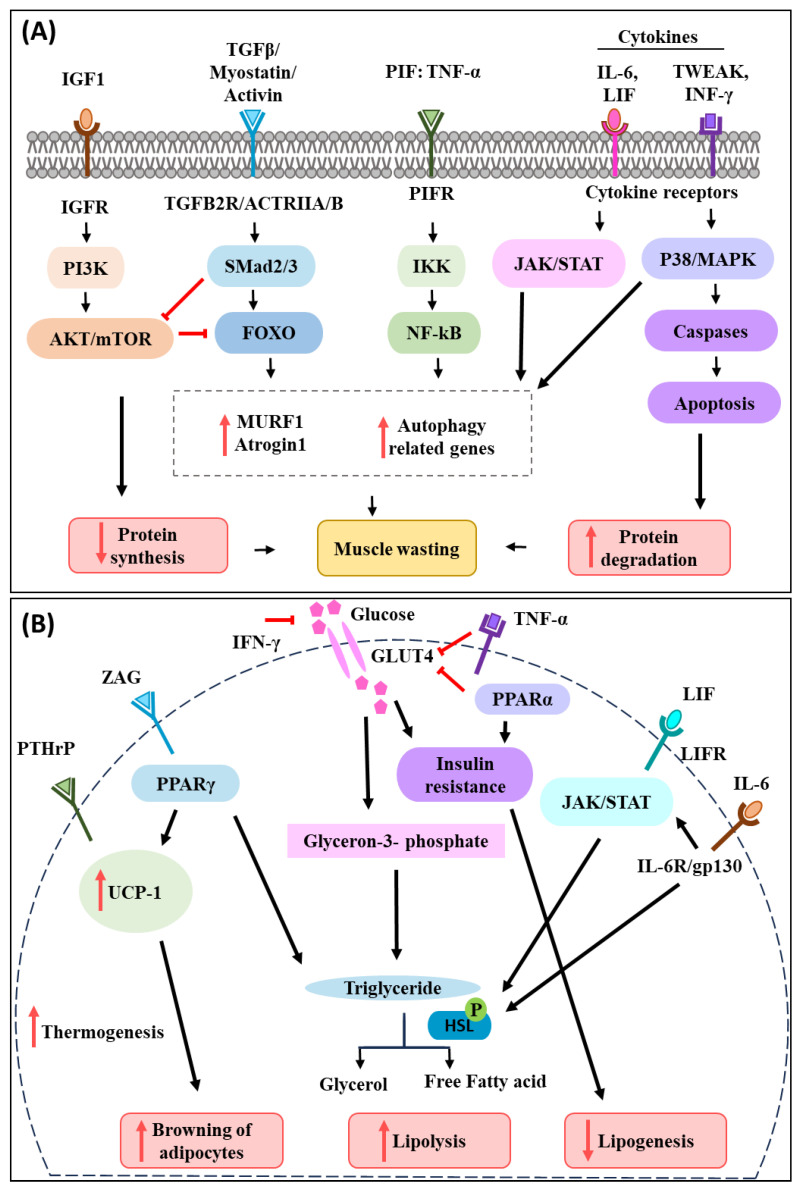
Schematic diagram of the most common signalling pathways active in muscle and adipocytes during cancer-associated cachexia. (**A**) During cancer-associated cachexia, several pro-inflammatory cytokines are secreted by the cancer cells. TWEAK, IFN-γ, TNF-α, IL-6, and LIF were found to enhance muscle catabolism via the p38/MAPK, NF-κB, and JAK/STAT pathways. TGFβ and members of its superfamily, including myostatin and activin, have been shown to induce muscle atrophy by stimulating FOXO, which in turn activates proteosome pathways. IGF-1 tends to trigger PI3K-AKT and mTOR signalling in muscle, enhancing protein synthesis. However, IGF-1 levels reduce during cancer-associated cachexia, which in turn affects muscle protein turnover and contributes to muscle atrophy. (**B**) Inflammatory cytokines, LIF, ZAG, and PTHrP released by the tumour cells also participate in adipose tissue wasting during cancer-induced cachexia. TNF-α mediates adipose wasting by decreasing GLUT4 expression, which results in inhibition of glucose transport and lipogenesis. Similarly, IFN-γ and PPARα stimulate lipolysis by inducing insulin resistance through recduction in glucose uptake. IL-6 and LIF were found to induce adipose lipolysis by activating the JAK/STAT signalling pathway, which causes HSL phosphorylation and therefore fat degradation. PTHrP activates UCP-1 in adipocytes, which results in browning of adipose tissue, and phosphorylation of HSL, which causes lipolysis. ZAG acts as a lipid mobilizing factor and stimulates adipose tissue wasting. Overall, inflammatory cytokines and cachectic factors released by the cancer cells activate various signalling pathways that lead to muscle and adipose wasting and together contribute to body weight loss during cancer-associated cachexia.

**Figure 2 cells-12-02598-f002:**
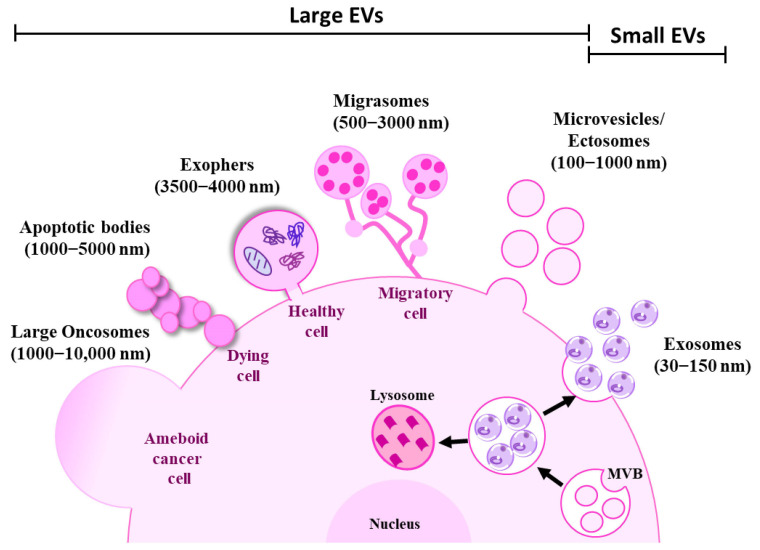
Schematic representation of extracellular vesicle subtypes. Cells are known to release heterogeneous populations of EVs with overlapping sizes. Small EVs such as exosomes (30–100 nm) are derived from intracellular endosomal compartments. ILVs are formed within MVBs which are subsequently released upon fusion with the plasma membrane as exosomes. Microvesicles or ectosomes (100–1000 nm) are larger in size and are directly budded off from the plasma membrane. Large EVs such as exophers (3500–4000 nm) are large membrane-bound vesicles that maintain homoeostasis of the cells by removing cellular waste, protein aggregates, and damaged organelles. Migrasomes (500–3000 nm) are another type of large EV released by the migrating cells as they move. Apoptotic bodies (1000–5000 nm) are produced by apoptotic or dying cells. On the other hand, micrometre-sized large oncosomes (1000–10,000 nm) are secreted from the membrane bleb of cancer cells.

**Figure 3 cells-12-02598-f003:**
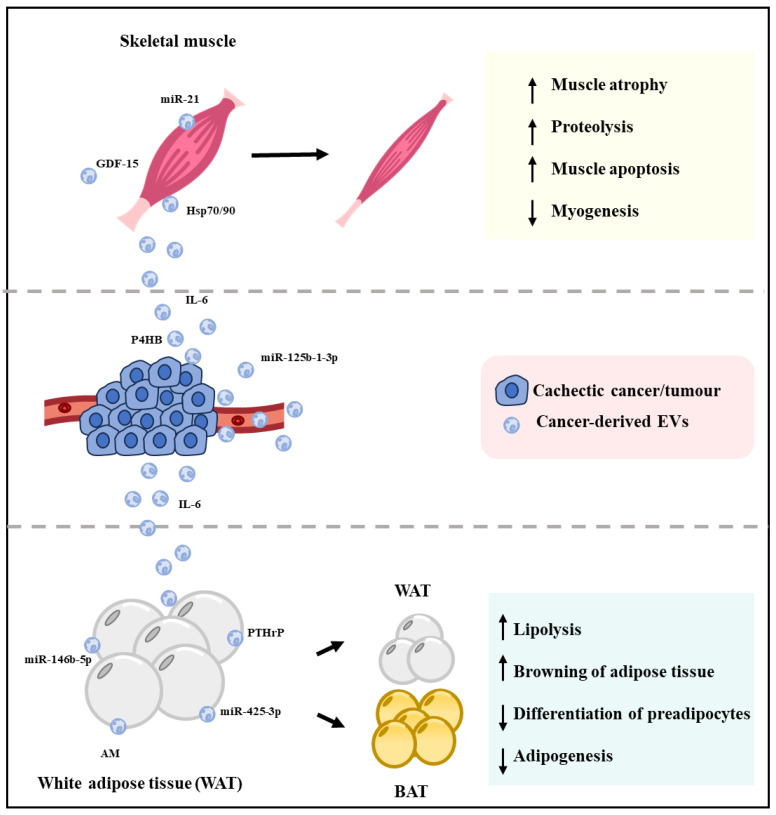
Cachexia induced by cancer-derived EVs. When EVs secreted by cachectic cancer cells are taken up by the muscle cells, it can promote muscle atrophy by decreasing myogenesis and increasing muscle apoptosis and proteolysis. Similarly, cachectic cancer-derived EVs can decrease adipogenesis and preadipocyte differentiation and increase lipolysis and browning of white adipose tissue, which together results in loss of adipose tissue. The resultant loss of muscle mass and adipocytes leads to loss of whole-body weight, which is known as cancer-associated cachexia.

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
