# Peer review of "Unravelling the Role of Cancer Cell-Derived Extracellular Vesicles in Muscle Atrophy, Lipolysis, and Cancer-Associated Cachexia"

_cells, 2023, doi:10.3390/cells12222598_

Round 1

Reviewer 1 Report

Comments and Suggestions for Authors

In the present review, Marzan and Chitti have summarized the functional role of tumor-derived extracellular vesicles (EVs) in the induction of cancer cachexia, with a particular emphasis on muscle wasting and lipolysis of white adipose tissue. This review is both timely and thorough, making it a valuable resource for readers engaged in the research field of EVs and cancer cachexia. However, I would like to emphasize that in the section addressing "Skeletal muscle wasting during cancer-associated cachexia," where the authors have already highlighted the role of tumor-induced suppression of myogenesis in cachexia development, as illustrated in the model figure, it's important to also acknowledge the contribution of other significant research findings pertaining to cancer-induced muscle regeneration. These findings are essential to provide a more holistic understanding of the complex interplay in the context of cancer cachexia. Pertinent references for these findings can be found as follows: PMID: 11009425; PMID: 29153940; PMID: 29875463.

Author Response

Dear Reviewer,

Thank you very much for taking the time to review this manuscript. Please find the attached detailed response below and the corresponding corrections highlighted in track changes in the re-submitted file. We are grateful for the constructive comments and we believe that the updated version is significantly improved.

Comment 1: In the present review, Marzan and Chitti have summarized the functional role of tumor-derived extracellular vesicles (EVs) in the induction of cancer cachexia, with a particular emphasis on muscle wasting and lipolysis of white adipose tissue. This review is both timely and thorough, making it a valuable resource for readers engaged in the research field of EVs and cancer cachexia. However, I would like to emphasize that in the section addressing "Skeletal muscle wasting during cancer-associated cachexia," where the authors have already highlighted the role of tumor-induced suppression of myogenesis in cachexia development, as illustrated in the model figure, it's important to also acknowledge the contribution of other significant research findings pertaining to cancer-induced muscle regeneration. These findings are essential to provide a more holistic understanding of the complex interplay in the context of cancer cachexia. Pertinent references for these findings can be found as follows: PMID: 11009425; PMID: 29153940; PMID: 29875463

Author response: We thank the reviewer for a wonderful suggestion. The point noted by the reviewer is indeed valid. As per the reviewer’s suggestion, we have now included a section (page 4, lines 129-147) discussing findings pertaining to cancer-induced muscle regeneration. 

Of note, along with muscle catabolism, tumour secreted factors also impair muscle stem cell function which results in the reduction of the regenerative ability of skeletal muscle [31, 32]. Hogan and colleagues have reported tumour-derived chemokine ligand 1 (CXCL1) suppresses myogenesis and alters the skeletal muscle immune microenvironment which together contributes to cancer-associated muscle wasting [33]. CXCL1 has particularly been found to inhibit muscle cell differentiation by promoting satellite cell proliferation and antagonizing cell cycle exit. In addition, CXCL1 was also found to stimulate the expansion of neutrophil-macrophage in skeletal muscle which together leads to impairment of regenerative ability of muscle cells [33]. Cytokine-induced degeneration of muscle cells is also known to partake in overall muscle wasting during cancer-associated cachexia [34, 35]. The activation of NF-κB by the cytokine TNF-α has also been shown to participate in inhibiting skeletal muscle differentiation by suppressing MyoD mRNA at the posttranscriptional level. Suppression of MyoD expression in muscle inhibits the formation of new myofibers and causes the degeneration of newly formed myotubes [34]. TNF-α and TGF-β cytokines have also been reported to upregulate metal-ion transporter ZRT- and IRT-like protein 14 (ZIP14) in the cachectic muscle which obstructs muscle-cell differentiation. ZIP14-mediated zinc accumulation in progenitor and differentiated muscle cells results in repression of the expression of MyoD, Mef2, and induces loss of myosin heavy chain [35].

Reviewer 2 Report

Comments and Suggestions for Authors As this is a review paper, I read it and commented negatively on the language but positively on the subject and the material covered. I do not see anything being missing from the material covered and aesthetically the graphs are fine. As is this being unfortunately poorly written in the linguistic side and that was my main concern. I bet there will be an opportunity to make minor revisions in the next round if the authors manage to improve the language (I tried to be nice, but the language is VERY problematic, and I am not sure the authors will address this part successfully in their resubmission). The review thematic structure on the other hand is fine and since original research is not included, I cannot comment on experimental work. As a theme it is unique (I have not seen a similar review on this subject) and important.

After major editing this interesting review can be published. The subject is of great interest to the medical community and biologists working with exosomes. 

Comments on the Quality of English Language

The manuscript contains numerous errors in the English language and requires major editing.

Author Response

Dear Reviewer,

Thank you very much for taking the time to review this manuscript. Please find the detailed response below and the corresponding corrections in track changes in the re-submitted files. 

Comment: As this is a review paper, I read it and commented negatively on the language but positively on the subject and the material covered. I do not see anything missing from the material covered and aesthetically the graphs are fine. As is this being unfortunately poorly written in the linguistic side and that was my main concern. I bet there will be an opportunity to make minor revisions in the next round if the authors manage to improve the language (I tried to be nice, but the language is VERY problematic, and I am not sure the authors will address this part successfully in their resubmission). The review thematic structure on the other hand is fine and since original research is not included, I cannot comment on experimental work. As a theme it is unique (I have not seen a similar review on this subject) and important.

After major editing this interesting review can be published. The subject is of great interest to the medical community and biologists working with exosomes. 

Author response: We thank the reviewer for the constructive feedback. As per the reviewer’s suggestion, we have tried our best to improve the linguistic side of the review. We believe that the updated version is significantly improved.